# Use of the Split Luciferase Complementation Assay to Identify Novel Small Molecules That Disrupt Essential Protein–Protein Interactions of Viruses

**DOI:** 10.3390/biom15121712

**Published:** 2025-12-09

**Authors:** Tisa Biswas, Richard E. Sutton

**Affiliations:** 1Section of Infectious Diseases, Department of Internal Medicine, Yale University, New Haven, CT 06520, USA; tisa.biswas@yale.edu; 2Department of Microbial Pathogenesis, Yale School of Medicine, New Haven, CT 06520, USA

**Keywords:** split luciferase complementation assay, viral protein-protein interactions, high-throughput screening, antiviral therapeutics, protein multimers, viruses

## Abstract

Protein–protein interactions (PPIs) are fundamental to viral replication, regulating transcription, assembly, and genome packaging. Despite their biological importance, few FDA-approved therapeutics directly target these complexes. The split luciferase complementation assay (SLCA) is a quantitative bioluminescence system to measure protein–protein interactions in vitro after the proteins in question have been fused in-frame to N and C luciferase fragments. The SLCA can be performed both in vitro using purified protein components and in live cells, as the luciferase substrate luciferin is cell-permeable, allowing detection of protein interactions in intact cells. Assay performance, however, depends on the expression level and stability of the fusion proteins used. SLCA has been successfully applied to target Rev–Rev interactions in human immunodeficiency virus type 1 (HIV-1) for high-throughput small-molecule screening, establishing a proof-of-concept to target other parts of the viral life cycle. The system can be extended to other pathogens that currently do not have specific antiviral therapies such as HIV-1 Tat–cyclin T1, Capsid dimerization in Dengue virus, capsid interactions in equine encephalitis viruses, capsid assembly in Epstein–Barr virus, and nucleoprotein oligomerization in rabies virus. These applications demonstrate how the assay’s ability to quantify multimeric structural interactions is essential to viral replication, providing an avenue to identify small-molecule inhibitors that prevent viral replication and spread. Although there are challenges to protein stability and assay optimization, the sensitivity and adaptability of the SLCA has broader implications in virology to accelerate antiviral drug development.

## 1. Introduction

Over the past century, outbreaks of the human immunodeficiency virus-1 (HIV-1), Dengue virus, Ebola virus, and, most recently, severe acute respiratory syndrome coronavirus 2 (SARS-CoV-2) have demonstrated both the speed and scale at which viruses can spread, becoming pandemics and affecting hundreds of millions of people. These viruses are borderless; they are not exclusive to developing countries but affect human populations worldwide, raising concerns for global health security. Dengue virus alone infects about 400 million people annually, and HIV-1 continues to impact more than 39 million individuals across the globe [1,2]. Although specific treatments exist for some of these infectious diseases such as HIV and influenza, they are often limited in efficacy and accessibility, leading to high morbidity rates. Importantly, many of these viruses lack specific therapies that directly target the pathogen. These challenges highlight the urgent need to identify new therapeutic strategies that can directly impact protein–protein interactions to prevent viral replication and prevent the virus from spreading.

Viruses rely heavily on the molecular machinery of host cells to replicate and spread throughout the body. This process, also known as obligate intracellular parasitism, is often mediated by protein–protein interactions between the viral protein and host factors. Once the virus enters the host cell, the protein–protein interactions drive viral transcription and replication. For instance, HIV-1 Tat binds to human protein cyclin T1 to initiate transcriptional elongation, while flaviviruses such as Dengue rely on capsid–capsid interactions to form a higher-order, multimeric structure [3,4]. These interactions are specific and critical to viral replication, highlighting the associated proteins as potential antiviral targets.

A proven method to target these interactions can be accomplished using the split luciferase complementation assay (SLCA), which is designed based on bioluminescence by reconstituting a functional firefly luciferase enzyme (~62 kDa in size) [5]. Luciferase, originally found in *Photinus pyralis* fireflies, is a class of enzymes that catalyze bioluminescent reactions across various organisms—including click beetles, lantern fish, and marine ostracod *Vargula hilgendorfii*, also known as the sea firefly, where the enzyme functions in processes such as mating, communication, defense, camouflage, and predation [6]. The SLCA specifically uses firefly luciferase, which is divided into two inactive fragments: the N-terminal (Nluc) and the C-terminal (Cluc). The firefly luciferase enzyme can be successfully bisected as follows: Nluc (1–416 amino acids) and Cluc (398–500 amino acids) [7]. The NLuc component is fused in-frame to one protein of interest, while the Cluc is fused to the other. When the two proteins interact in the assay, it brings the luciferase fragments in close enough proximity to form an active enzyme (Figure 1). This reconstituted enzyme then generates a quantifiable bioluminescent signal in the presence of ATP and luciferin; ATP is poorly cell-permeable due to its hydrophilic nature, whereas luciferin is cell-permeable [7]. This system is used to detect interactions between structural proteins that multimerize, precisely quantifying protein–protein interactions that are critical to viral replication.

The SLCA can be performed both in vivo and in vitro, allowing quantitative detection of protein–protein interactions in live cells, cell lysates, or purified proteins [8,9,10]. In mammalian cell systems, constructs encoding NLuc- and CLuc-tagged proteins can be introduced either by co-transfection or by separate transfection, followed by the subsequent mixing of lysates. HEK293T cells are commonly used due to their robust protein expression and high signal-to-noise ratios, and the assay can be applied to other cell lines with adequate protein expression [10].

Adaptations of this system have extended its application to animal models. Specific fragment combinations of firefly luciferase—Nfluc(1–475)/Cfluc(245–550), Nfluc(1–475)/Cfluc(265–550), and Nfluc(1–475)/Cfluc(300–550)—produced detectable luminescence in subcutaneous 293T cell implants in mice upon addition of d-luciferin, recovering 0.01–4% of the activity observed with intact luciferase [11]. Delivery of one fragment (TAT–Cfluc(265–550)) to cells stably expressing the other fragment (Nfluc(1–475)) restored luminescence, demonstrating the assay’s potential for monitoring molecular interactions in vivo [11].

In addition to the SLCA, other luciferase complementation systems have been developed, such as Renilla luciferase assay, that can detect protein–protein interactions by the reconstitution of the bioluminescent enzyme derived from the sea pansy (Renilla reniformis). This method has been commonly used to measure transient protein–protein interactions inside mammalian cells and plant protoplasts [12]. Unlike firefly luciferase, Renilla luciferase is a smaller enzyme (~37 kDa in size) that catalyzes the oxidation of coelenterazine (cell-permeable) to generate bioluminescence [13]. Beyond the luciferase-based systems, non-luciferase complementation methods include the split green fluorescent protein (GFP), which is often used to study stable protein–protein interactions or for sub-cellular imaging. In split GFP assays, a similar process occurs as two non-fluorescent proteins are fused to different parts of GFP, and GFP fluorescence is restored when the proteins come into close proximity by the protein–protein interactions. While the SLCA can detect real-time readouts of the bioluminescence signal that helps to track when interactions form and break, the inactive GFP fragments in the split GFP assay can irreversibly fold into a complete beta barrel upon interaction and produce a fluorescent signal, even if the proteins later separate [14,15]. Similarly to the split GFP system, the split alkaline phosphatase or AP assay produces an irreversible signal after protein–protein interactions reconstitute a functional AP that cleaves phosphate groups from the added substrates (such as the hydrophilic p-nitrophenyl phosphate or hydroquinone diphosphate) to generate the visible signal in the form of color change or fluorescence [16,17]. This is particularly useful when studying protein interactions that occur on the cell surface or even extracellularly.

Specifically, the SLCA has several advantages over biochemical assays currently used to study protein–protein interactions, such as co-immunoprecipitation (co-IP), mass spectrometry (MS), and surface plasmon resonance (SPR). The SLCA can be used to study protein–protein interactions both in vivo and in vitro, which provides flexibility for the types of experiments being conducted. Additionally, when SLCA is performed in live cells, the two proteins of interest are fused to luciferase fragments within the cellular environment, allowing detection of low-affinity interactions that are often lost during lysis and immobilization processes required for Co-IP and MS [18,19]. This system is also well-suited for high-throughput screening that can rapidly test hundreds of thousands of small molecules or compounds to identify new therapeutic candidates that either activate or inhibit the protein–protein interaction.

The potential applications of the SLCA in virology are broad. Viruses such as HIV-1, Dengue, Eastern equine encephalitis virus (EEEV), Western equine encephalitis virus (WEEV), Epstein–Barr virus (EBV), and Rabies virus rely on protein–protein interactions for viral replication and spread throughout the infected organism. The SLCA has been used to quantify Rev–Rev interactions in HIV-1 both in vitro and in vivo to optimize the assay for high-throughput screening to identify inhibitors of the interaction to prevent the export of intron-containing viral RNA from the nucleus and thus viral replication [20]. Beyond HIV-1, the utility of split luciferase-based assays for discovering modulators of viral protein interactions has been demonstrated in hepatitis B virus (HBV), where a cell-based split luciferase complementation (SLC) assay was used to monitor core protein dimerization. Using this system, Arbidol and 20-deoxyingenol were identified from a 672-compound library as regulators of HBV core dimerization, which correspondingly altered HBV DNA replication in vitro [21]. Although this method differs from SLCA used in HIV-1 and other viruses, it provides proof-of-concept that split luciferase complementation can be adapted to screen for inhibitors or modulators of critical viral protein–protein interactions in diverse viral systems. Many HIV-1 protein–protein interactions remain untargeted, such as Tat–cyclin T1 and nucleocapsid. Nucleocapsid interactions are necessary for viral transcription and replication. Together, these findings serve as a proof-of concept, demonstrating SLCA’s potential as a highly adaptable and impactful tool for the discovery of antiviral agents and for mechanistic studies of viral proteins to advance therapeutic development across multiple viral pathogens.

In this article, we outline how the SLCA method can be used to target viral protein–protein interactions across six globally widespread viruses that currently lack specific antiviral therapies. We present the successful application of the assay against HIV-1, where the quantification of the Rev–Rev interaction serves as a proof-of-concept for targeting additional interactions in the virus, as well as Dengue, Eastern/Western equine encephalitis, Epstein–Barr, and rabies viruses. By extending applications to other clinically relevant viruses, we discuss the potential of SLCA as a sensitive and adaptable platform, applicable to any virus with multimerizing protein–protein interactions, to conduct high-throughput screening to identify new antiviral drug candidates. To provide a framework for understanding the therapeutic potential of targeting viral protein–protein interactions, we include a brief overview of FDA-approved small molecules that modulate such interactions. These examples demonstrate that protein-protein interfaces are targets for drugs in both infectious diseases and other pathologies, providing clinical proof-of-concept for approaches like the SLCA to identify novel antiviral agents.

## 2. SLCA Applications in Viruses

### 2.1. Human Immunodeficiency Virus Type-1 (HIV-1)

HIV-1 is an ongoing pandemic, affecting nearly 40 million people worldwide [22]. While sub-Saharan Africa remains the most heavily impacted region, as it accounts for almost two-thirds of all global infections, HIV-1 persists in every region of the world, including high-income countries such as the United States, parts of Europe, and East Asia [23]. The persistence of HIV-1 as a global epidemic is attributable to its ability to integrate its genetic material within the host genome, primarily within CD4+ T cells and macrophages, and establish transcriptionally silent latent reservoirs that remain unaffected by current antiretroviral therapies (ART). As a result, the virus is never eradicated from an infected person, and treatments must be lifelong, as brief interruptions can lead to viral rebound and clinical disease [24]. Therefore, identifying new mechanisms to reverse HIV-1 latency or control viral replication is critical for advancing therapeutics and may ultimately impact the cure of infected individuals.

One approach to targeting viral protein–protein interactions is the SLCA that has been successfully used to study Rev–Rev interaction in HIV-1. Rev is a viral regulatory protein that multimerizes on the Rev Response Element (RRE) within the major intron to facilitate the nuclear export of the unspliced viral RNAs to the cytosol [20]. The Rev–Rev protein interaction occurs to form dimers and oligomers for efficient RRE binding [25]. If the Rev–Rev interaction is inhibited, then intron-containing viral RNA cannot be translated, and the virus cannot replicate. Application of the SLCA to quantify the Rev–Rev interaction both in vitro and in vivo demonstrated a high Z′ factor of 0.85, suggesting that the cell-free assay can be further used for the high-throughput screening of inhibitory small molecules [20]. This study serves as a proof-of-concept to detect and quantify HIV-1 regulatory protein and structural proteins to potentially inhibit HIV from transcribing and replicating in cells.

Capsid–capsid interactions can also be studied using the SLCA. The HIV-1 capsid protein (approximately 231 amino acids) multimerizes to form a protective shell around the diploid viral RNA genome, protecting it from degradation and facilitating reverse transcription to convert the viral RNA into DNA [26]. After reverse transcription, the nucleocapsid and capsid structure maintains the integrity of the viral core and interacts with host proteins to deliver the viral DNA into the nucleus and integrate the duplex viral DNA into the host genome. Inhibition of capsid–capsid interaction disrupts its multimerization process, preventing proper assembly and disassembly of the viral core and thereby blocking viral replication [26].

The FDA-approved capsid inhibitor Lenacapavir (LEN) targets capsid multimerization to suppress viral replication (Figure 2). Although LEN, an FDA-approved HIV-1 capsid inhibitor, was not discovered using the SLCA, it provides clinical proof-of-concept that disrupting viral protein–protein interactions is a viable antiviral strategy. Its mechanism of action—interfering with capsid assembly (when the capsid proteins produce a conical structure in newly formed virions within the cell) and disassembly (when the capsid disassembles and releases the viral genome within the nucleus of the host cell)—demonstrates the therapeutic potential of identifying small molecules that interfere with viral protein–protein interactions (Figure 3) [27]. SLCA-based high-throughput screening therefore provides a complementary strategy to identify small molecules capable of modulating critical viral interactions similar to those targeted by LEN. LEN is highly potent, with a mean half-maximal inhibitory concentration (IC-50) value of 200 pM for HIV-1 [28].

LEN binds at a conserved pocket formed between capsid subunits at the interface of the N-terminal domain of one monomer and the C-terminal domain of an adjacent subunit. The allosteric binding stabilizes the hexameric capsid lattice, inducing a hyperstable conformation that disrupts the normal metastability required for the HIV-1 life cycle [29]. During early infection, LEN prevents uncoating by hyper-stabilizing the capsid lattice, thereby blocking reverse transcription and nuclear import [29,30]. In later stages, it interferes with the proper assembly of Gag-derived capsid proteins, leading to the formation of morphologically defective virions [31]. These multifaceted actions collectively result in potent inhibition of viral replication at multiple steps of the life cycle.

Although LEN is highly effective, it does not fully inhibit all stages of capsid/nucleocapsid assembly and resistance to the drug can develop over time due to specific mutations in capsid, notably M66I. The M66I substitution is a single amino acid change in the capsid that interferes with LEN binding, reducing its potency by more than 3000-fold [32]. This mutation alters the drug-binding pocket without disrupting the capsid’s overall structure or ability to assemble, allowing the virus to remain infectious despite reduced drug susceptibility (Figure 4) [33]. This highlights the importance of discovering new antiretroviral drug candidates to inhibit capsid–capsid, nucleocapsid–nucleocapsid, or HIV-1 Tat–cyclin T1 interactions. A major barrier to curing HIV-1 is viral latency, allowing the virus to remain transcriptionally quiescent in infected immune cells and reactivate if treatment is discontinued. The viral regulatory protein Tat (Trans-Activator of Transcription) plays a central role in this reactivation by binding to the host transcription factor cyclin T1 to stimulate transcriptional elongation of the viral genome [34]. Disrupting this interaction would prevent RNA polymerase II from efficiently transcribing the provirus and prevent viral replication from spreading further. Despite its critical role in viral transcription, no FDA-approved therapies currently target Tat–cyclin T1 binding, and how the Tat–cyclin T1 interaction could be directly inhibited remains unexplored. As a first step in understanding this interaction and identifying potential therapeutic avenues, an SLCA could be used to quantitatively assess the stability and behavior of HIV-1 Tat–cyclin T1 interaction in vitro in order to then test known inhibitors that have been shown to inhibit Tat from binding to the trans-activation response or TAR RNA, such as Roscovitine and Gemcitabine, to validate the assay and conduct a high-throughput screen [35].

### 2.2. Dengue

Dengue virus is one of the most common mosquito-borne viruses in the world, yet there are no specific antiviral drugs that target this virus. When the virus infects human host cells, primarily macrophages and dendritic cells, it releases its positive-sense single-stranded RNA genome directly into the cytoplasm to function as an mRNA. This allows the host ribosomes to recognize the viral RNA and initiate translation into a single, non-functional polyprotein [36]. However, subsequent polypeptic enzymatic cleavage by viral NS3 protease, in conjunction with its cofactor NS2B, generates functionality, including structural proteins (capsid, prM/M, and E) and nonstructural proteins (NS1-NS5) [37]. Among the structural proteins, the capsid (C) protein is responsible for packaging the viral genome and forming a nucleocapsid core. The C protein is small (~100 amino acids) and has a high concentration of positively charged basic residues, which forms strong electrostatic interactions with the negatively charged phosphate backbone of the viral RNA [38]. For assembly, two C proteins first dimerize (C–C interaction), which serves as a building block for nucleocapsid formation. These C–C dimers then oligomerize around the viral RNA, resulting in a protective, multimeric nucleocapsid that is later enveloped by the prM/M and E proteins during virus maturation [3]. This physically shields the viral RNA from degradation by host nucleases and establishes a structural foundation for the production of infectious virions. If the C–C interaction can be quantified by the SLCA to subsequently identify potential inhibitors of that interaction, then the proteins cannot oligomerize to form the nucleocapsid and the viral RNA will be vulnerable to degradation, preventing the assembly of new virions to block viral replication [39,40].

### 2.3. Eastern Equine Encephalitis Virus (EEEV)

EEEV is a mosquito-borne alphavirus, mainly transmitted by Culiseta melanura, with no current medications that specifically target the virus. Upon entry in host cells—primarily dendritic cells, macrophages, and fibroblasts—the viral positive-sense single-stranded RNA genome is directly released into the cytoplasm, where it functions as mRNA for immediate translation [41]. The genome encodes structural proteins (C, E1, E2, E3, and 6K) that assemble into viral particles at the host cell’s plasma membrane [41]. Compared to other Western Hemisphere alphaviruses, such as Venezuelan equine encephalitis virus (VEEV), the capsid protein of EEEV can interact with the host transport machinery to exit the nucleus by binding to Importin α/β and CRM1 (exportin 1) to suppress RNA polymerase II-mediated transcription that the host cell uses to make mRNA for antiviral proteins [42]. By blocking the nuclear core complex, EEEV capsid prevents the nuclear import of transcription factors (e.g., STAT1, NF-κB) responsible for activating antiviral genes and the export of host mRNAs necessary for immune signaling and cellular function. Selective inhibitors of nuclear export (SINE) compounds such as KPT-185, KPT-335/verdinexor, and KPT-350 were found to bind to CRM-1 in VEEV, preventing the nuclear export of capsid [43]. This causes the capsid proteins to accumulate in the nucleus, which then prevents them from entering the cytoplasm to multimerize around the viral RNA to form the multimeric structure that protects the viral RNA (Figure 5) [9]. It is still unclear, however, whether SINE compounds directly inhibit the EEEV capsid–capsid interactions. This question can be addressed using the SLCA to validate previous findings and to identify more potent analogs of these compounds that can specifically target this protein–protein interaction.

### 2.4. Western Equine Encephalitis Virus (WEEV)

WEEV is another mosquito-borne alphavirus, generally less severe than EEEV, with no current medications that can target the virus. Similarly to EEEV, the virus is a positive-sense, single-stranded RNA virus. Its genome is released into the host cytoplasm where it directly acts as mRNA [44]. Viral entry depends on interactions between the envelope glycoproteins E1 and E2, which form heterodimers on the virion surface. For wild-type WEEV strains, these E2–E1 heterodimers create binding sites for host receptors, with PCDH10 binding in the cleft [45]. However, a single amino acid change in E2 (V265F) can cause a conformational change in the binding site, thereby broadening the range of cell types the virus can infect. In principle, inhibitors that target the E1–E2 interaction or the receptor interfaces could block viral entry into cells, and the SLCA system can be adapted to quantify the E2–E1 interactions to screen for new potential therapeutic compounds.

In addition to the E1 and E2 structural proteins, the genome also encodes capsid proteins that multimerize around the viral RNA in the cytoplasm to form nucleocapsids (Figure 5). Thieno [3,2-b] pyrrole-based inhibitors exhibit antiviral activity against neurotropic alphaviruses like WEEV in cultured cells [46]. The mechanisms by which the inhibitors block viral replication, however, are unclear at present. Based on the experimental approach with a replicon-based assay, it indicates that the drug interferes with the viral replication machinery rather than viral entry into cells [47]. Thus, the SLCA could be used to determine whether the thieno [3,2-b] pyrrole-based inhibitors can directly disrupt capsid–capsid interactions and to quantitatively assess their antiviral potency.

### 2.5. Epstein–Barr Virus (EBV)

EBV is a human herpesvirus that infects more than 90% of the world population [48]. The virus primarily targets B lymphocytes and oropharyngeal epithelial cells [48]. During infection, the virus binds to the CD21 receptor on B cells via its glycoprotein gp350/220 or enters epithelial cells through gH/gL- and gB-mediated endocytosis, releasing its viral double-stranded DNA genome into the nucleus where early transcription takes place [49]. The genome encodes structural proteins such as the major capsid protein BcLF1, which binds to other BcLF1 monomers to form the hexons and pentons of the icosahedral lattice (Figure 5). The capsid is further stabilized by triplexes that are formed by minor capsid proteins, BORF1 (Tri1), and BDLF1 (Tri2) that bridge BcLF1 units [50]. Together, these interactions assemble a stable nucleocapsid that protects the viral genome from degradation. If the BcLF1–BcLF1 or BcLF1–triplex interaction can be targeted to prevent capsid assembly, it could inhibit virion maturation, which could then prevent the production of infectious EBV particles [50]. While there are no clinically approved inhibitors targeting capsid assembly or viral DNA synthesis, spironolactone, a mineralocorticoid receptor antagonist, has been shown to inhibit the function of the EBV SM protein, which is critical for late lytic gene expression and capsid antigen production [51]. This demonstrates that small molecules can interfere with capsid formation and virion maturation. The SLCA can provide a platform to test whether spironolactone can also target the BcLF1–BcLF1 or BcLF1–triplex interactions and screen additional inhibitors of the interactions.

### 2.6. Rabies Virus

Rabies is a negative-sense single-stranded RNA virus primarily transmitted to humans through saliva from an infected animal with no specific antiviral treatments currently available, and death is invariable. When rabies virus infects host neurons, the viral RNA genome is released into the cytoplasm, where it serves as a template for transcription and replication by the viral RNA polymerase [52]. The genome encodes structural proteins including nucleoprotein (N), phosphoprotein (P), matrix protein (M), glycoprotein (G), and the large polymerase (L) protein. The N protein (~450 amino acids) multimerizes around the viral RNA to form a ribonucleoprotein (RNP) complex that not only protects the viral genome from the host RNAses but also provides the stability and structure for the microtubule-based retrograde transport (Figure 5) [53]. After initial viral replication near the bite site, rabies virions are transported from the peripheral nerve endings to the neural cell body in the spinal cord and brain. The ribonucleoprotein or RNP complexes and virions hijack the host’s microtubule network and engage dynein motor proteins to directionally travel toward the minus end of microtubules, which transports the viral genome to the neuronal nucleus–proximal region, where transcription, replication, and assembly occur [54]. Without this directional transport, the virus would remain trapped at distal nerve terminals and would not be able to invade the CNS. The N–N protein interaction is critical for the virus to spread, and the SLCA can be used to identify small-molecule inhibitors that disrupt N oligomerization to destabilize the RNP complex and prevent viral replication and the assembly of infectious virions [55].

## 3. SLCA

To identify inhibitors of the specific protein–protein interaction for the above applications, the protein nucleic acid sequences are fused in-frame to complementary luciferase fragments. Homotypic interactions, where identical proteins interact (e.g., capsid–capsid or Rev–Rev), often produce strong and consistent bioluminescence [18]. However, heterotypic interactions, which involve two different proteins, may require optimization, as one protein is typically fused to the NLuc fragment and the other to the CLuc fragment, or vice versa [18]. The fusion orientation of target proteins to luciferase fragments can influence assay performance by affecting protein folding or the accessibility of interaction interfaces. In such cases, empirical testing may be required to identify optimal configurations. The optimal amino acid length for each protein in the assay is not fully established; however, based on prior studies, proteins of approximately 200–300 amino acids generally perform well in the SLCA.

The reconstitution of luciferase activity occurs only when the two proteins interact, providing a quantitative readout in relative light units (RLUs). The assay is first optimized and validated using positive and negative controls, with a Z′ or Z factor of between 0.5 and 1.0, indicating sufficient robustness for high-throughput screening [56]. The presence of a well-characterized inhibitor produces a ≥5-fold reduction in RLU. It is always quite helpful to have an already-identified inhibitor to reduce RLU by at least 5-fold (termed an S/B value), which can improve the reliability of the assay during optimization (Figure 6).

Once validated with acceptable Z′ and S/B values, the system can be used to test small-molecule inhibitors to target the protein–protein interface, scaled to a 384-well format for high-throughput screening using an exceptionally complex library of compounds [57]. A reduction in RLU indicates a potential disruption of the protein–protein interaction, and the extent of inhibition can be quantified through dose–response analysis by testing different concentrations of the compounds.

Compounds identified as preliminary hits should be tested against the full-length firefly luciferase (FFLUC) to eliminate factors that can directly inhibit the luciferase enzyme. Verified hits can be further evaluated using biochemical assays, and confirmed hits can be optimized chemically to increase potency and selectivity. Only compounds that demonstrate specific and reproducible inhibition of the protein–protein interaction and functional antiviral activity advance to preclinical development. Subsequent cell- and animal-based assays are used to quantify antiviral activity and verify that the compounds are functional, prior to any clinical studies.

## 4. FDA-Approved Inhibitors

Several FDA-approved small molecules have established that protein–protein interactions are viable therapeutical targets, providing a strong rationale for developing novel antivirals using the SLCA. In virology, compounds such as Fostemsavir and Maraviroc exemplify this approach by inhibiting interactions between the HIV-1 envelope glycoprotein gp120 and host receptors CD4 and CCR5, thereby blocking viral entry into host cells [58]. LEN extends this concept by targeting the viral capsid–capsid interaction, demonstrating that the disruption of structural protein multimerization impedes capsid assembly and consequently inhibits viral replication [28]. LEN is now FDA-approved for both HIV-1 treatment and prophylaxis [32]. Beyond virology, the success of PPI-targeting agents in oncology and immunotherapy further underscores the translational potential of this approach (Table 1). Venetoclax, for example, disrupts BCL2–BCL2 interactions to induce apoptosis in malignant cells [59], whereas Motixafortide antagonizes the CXCL12–CXCR4 axis to mobilize hematopoietic stem cells for the treatment of multiple myeloma [60]. These FDA-approved compounds span diverse mechanisms, demonstrating that protein–protein interactions can be targeted at multiple stages, including viral entry, replication, assembly, and immune modulation. However, gaps remain regarding the type of diseases these drugs target (mostly cancer), potency, and resistance to the medication over time, highlighting the importance of continued research and utilizing the SLCA to identify new drugs developed for deadly infectious diseases that are currently not targeted.

## 5. Conclusions

Despite advances in antiviral research, many clinically important viruses such as HIV-1, Dengue, equine encephalitis viruses, Epstein–Barr virus, and rabies remain major global health threats. Although there are current antiretroviral therapies for HIV-1, there is no cure for those infected with HIV-1, mainly due to viral latency. Other viruses such as Dengue, equine encephalitis viruses, EBV, and rabies do not have specific antiviral therapies, which can lead to severe acute disease and persistent virologic presence, depending on the viral infection. These limitations highlight the importance of identifying novel therapeutic candidates by using known methods such as the SLCA to target viral processes at the molecular level. The SLCA provides quantitative results both cell-free and within cells, with high sensitivity and low background that is adaptable to studying a number of viral protein–protein interactions. The system is best suited for detecting intracellular protein–protein interactions, and its applications to extracellular or membrane-bound targets has not yet been demonstrated in the literature. By demonstrating its applications across multiple viruses, including HIV-1 as a proof-of-concept, this article provides a framework for future research aimed at developing antiviral therapies for viruses that currently lack specific therapies. A poignant example of this is LEN, which inhibits both virus disassembly and assembly in cells, and is now FDA-approved for treatment and prophylaxis of HIV-1.

## 6. Future Perspective

With many millions of individuals worldwide affected by HIV-1, Dengue, eastern and western equine encephalitis, Epstein–Barr, and rabies viruses, the next frontier in antiviral research may lie in targeting viral multimeric structural protein interactions that drive viral replication and cause human disease. Protein–protein interactions, such as HIV Tat–cyclin T1, Dengue capsid dimers, and rabies nucleocapsid oligomers, are highly specific and can be targeted with small-molecule inhibitors. To identify such inhibitors, the SLCA or a related assay can be utilized to conduct high-throughput screening of hundreds of thousands or millions of small molecules. With emerging viruses on the rise, SLCA reliably and reproducibly quantifies protein–protein interactions at various stages within the viral cycle and should thus accelerate treatment and possibly cure of acute or chronic viral infections. Importantly, therapies developed through this approach could provide affordable treatment options for patients throughout the world, in low- and middle-income countries in South America, Africa, and Asia.

## Figures and Tables

**Figure 1 biomolecules-15-01712-f001:**
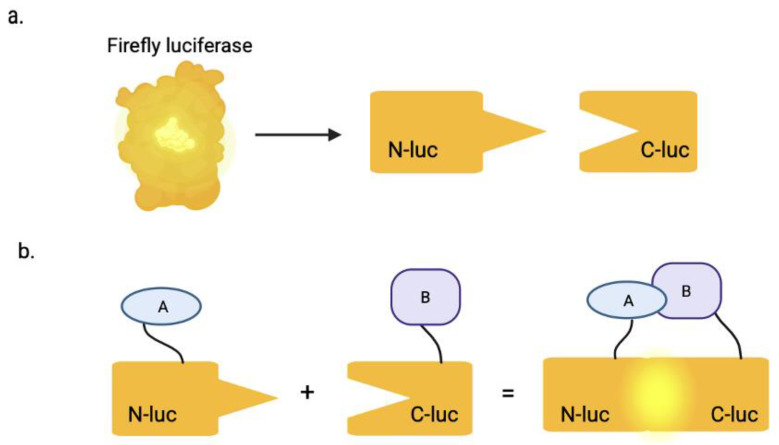
The split luciferase complementation assay. (**a**) The full-length luciferase enzyme is split into two inactive fragments: the N-terminal (Nluc) and C-terminal (Cluc). (**b**) The Nluc is fused in-frame to protein A and Cluc is fused to protein B. When protein A binds to or interacts with protein B, it brings the two inactive enzyme fragments in close proximity, which results in a bioluminescent signal, after addition of ATP and luciferin. Created in BioRender. Biswas, T. (2025) https://BioRender.com/iexkcts.

**Figure 2 biomolecules-15-01712-f002:**
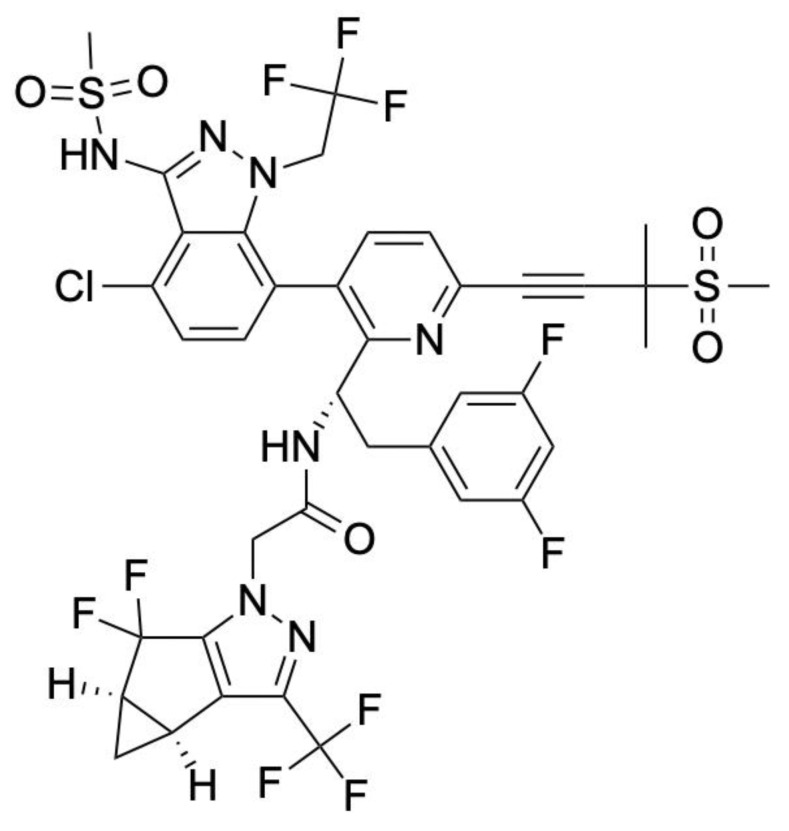
Chemical structure of Lenacapavir. LEN contains a tricyclic core that facilitates high-affinity binding at the interface of HIV-1 capsid hexamers, blocking the assembly and disassembly of the viral core. Its side groups form stable hydrogen-bonding and hydrophobic interactions with the capsid protein, that ultimately contributes to its strong antiviral activity even at a low pM concentration.

**Figure 3 biomolecules-15-01712-f003:**
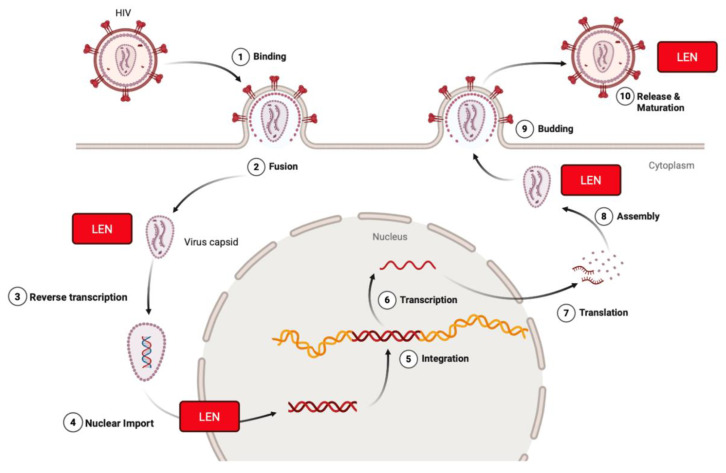
Lenacapavir inhibits HIV-1 at multiple stages of the viral life cycle. Lenacapavir (LEN) binds to the HIV-1 capsid protein (p24) at the interface of hexamer subunits. It blocks nuclear import of viral DNA by preventing interaction with host nuclear transport proteins, disrupts proper capsid core formation by altering subunit assembly after translation, interferes with Gag/Gag-Pol function to inhibit virus assembly and release, and can interact with HIV once it is released from the plasma membrane during its process of maturation. Created in BioRender. Biswas, T. (2025) https://BioRender.com/hjzop1g.

**Figure 4 biomolecules-15-01712-f004:**
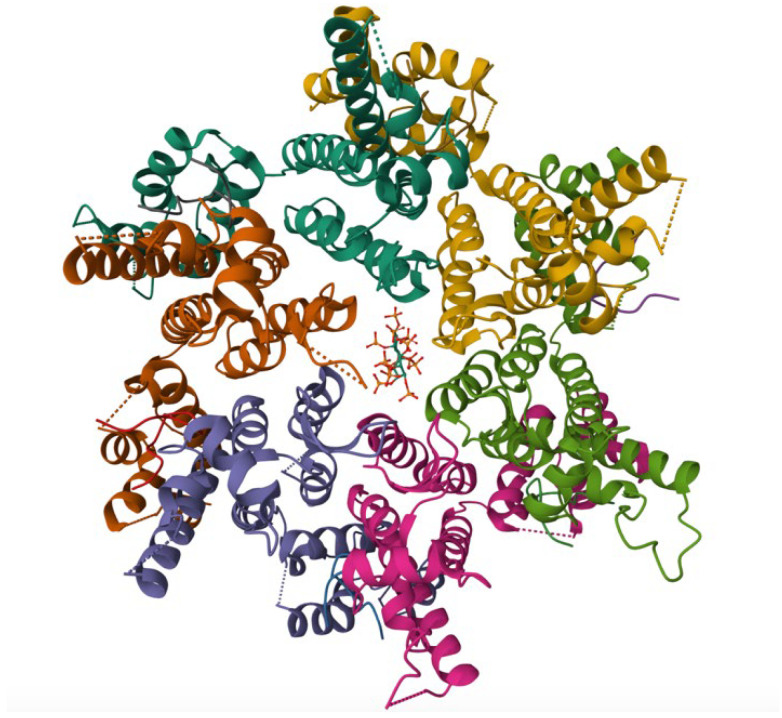
Crystal structure of the M66I HIV-1 capsid hexamer stabilized by disulfide bonds in complex with the human CPSF6 peptide and IP6. The protein, expressed in *E. coli*, highlights the M66I resistance mutation within the capsid protein, providing insight into the molecular basis of reduced LEN binding while maintaining capsid structural integrity. Structural coordinates were obtained from PDB entry 8GDV [33].

**Figure 5 biomolecules-15-01712-f005:**
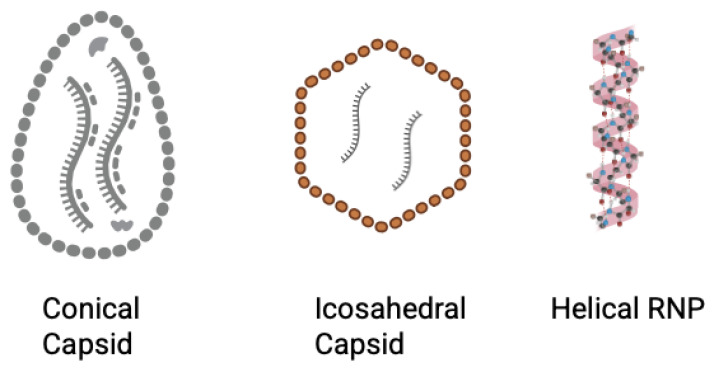
Schematic of capsid structures across the six viruses. HIV-1 capsid multimerizes and forms a conical structure (left). Dengue virus has an icosahedral outer surface (represented by the red outer layer), while EBV forms an icosahedral capsid, and EEEV and WEEV forms an icosahedral nucleocapsid that encompasses the viral genome (center). Rabies virus forms a helical RNP complex that functions as a nucleocapsid. (right). Created in BioRender. Biswas, T. (2025) https://BioRender.com/f2cjozz.

**Figure 6 biomolecules-15-01712-f006:**
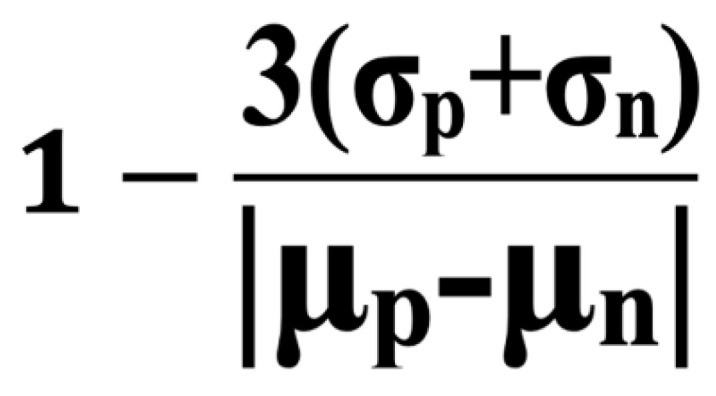
Formula of Z′. The Z′ (Z factor) is a statistical parameter to evaluate the quality of high-throughput screening assays. It is calculated using the means (μ) and standard deviations (σ) of the positive (p) and negative (n) control signals. S/B is μ*_p_* divided by μ*_n_*.

**Table 1 biomolecules-15-01712-t001:** FDA-approved inhibitors that disrupt protein–protein interactions.

Compound	Target	Binding Interaction	Disease or Virus	Mechanism of Action
Fostemsavir	Viral gp120	Viral gp120-CD4	HIV-1	Blocks viral gp120 from binding to human CD4 receptor, preventing viral entry
Maraviroc	CCR5 co-receptor	Viral gp120-CCR5	HIV-1	Inhibits R5-tropic viral gp120 from binding to CCR5 co-receptor, blocking viral entry
Lenacapavir	Capsid	Capsid–Capsid	HIV-1	Disrupts capsid assembly and disassembly at multiple stages within the viral cycle
Venetoclax	BCL2	BCL2-BIM	Chronic lymphocytic leukemia, Small lymphocytic leukemia, Acute Myeloid Leukemia	Disrupts anti-apoptotic protein BCL2 and pro-apoptotic protein BIM to induce apoptosis
Motixafortide	CXCR4	CXCL12-CXCR4	Multiple myeloma	Blocks CXCR4 receptor from binding to its ligand CXCL12, mobilizing hematopoietic stem cells from the bone marrow

## Data Availability

Not applicable.

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
