# Peer review of "Use of the Split Luciferase Complementation Assay to Identify Novel Small Molecules That Disrupt Essential Protein–Protein Interactions of Viruses"

_biomolecules, 2025, doi:10.3390/biom15121712_

Round 1

Reviewer 1 Report

Comments and Suggestions for Authors

This is an excellent review of the SLCA. It is an interesting assay that will be useful for many drug screening assays.  The review may be useful for anyone looking for a high throughput assay to identify inhibitors of a protein:protein interaction.  The article is well written and informative.  It is interesting that it reviews a broad range of viruses.

Specific points

  1. How does that assay work in live cell screening? Is the luciferase substrate cell permeable? Does it work for all cells?  How does the assay work in vitro using purified components?
  2. It would be helpful to show some actual data obtained with the assay. Do the authors have some data to make a figure or perhaps get permission to use some published data from the assay. 
  3. What amino acid residues constitute the A and B luciferase fragments? What are the sizes of the fragments. How was it decided where to split the protein?  
  4. Table 1 has a column labeled “target”. This shows the protein interaction that is being targeted.  It does not state the actual target of the drug.  An additional column should be added that lists the target of the drug.
  5. The authors refer to a list of anti-cancer mAbs (Atezolizumab, etc.) as evidence that drugs can target protein:protein interactions. These are all mAbs.  Would the SLCA be suitable to find drugs that are mAbs?  Could the assay be used to screen for drugs against cell surface receptors?  It would be helpful to state which type of protein:protein interactions can be targeted. 
  6. Some description of the enzyme substrates used in the assay should be added.
  7. It would seem that the assay might be dependent on the site at which the A and B proteins are fused to the N-Luc and C-Luc fragments. Does it matter?  Does it always work?
  8. A few minor grammatical errors to correct. Page 3 third paragraph should be “are broad” rather than “is broad”.  Also cite reference for the Rev:Rev assay screen in that paragraph.  Page 6 first paragraph should be “that have” not “that has”.
  9. Has anyone tested LEN in SLCA? Does it work?  LEN might not inhibit CA:CA interaction but might stabilize it. 

Author Response

This is an excellent review of the SLCA. It is an interesting assay that will be useful for many drug screening assays.  The review may be useful for anyone looking for a high throughput assay to identify inhibitors of a protein:protein interaction.  The article is well written and informative.  It is interesting that it reviews a broad range of viruses.

Specific points

1. How does that assay work in live cell screening? Is the luciferase substrate cell permeable? Does it work for all cells?  How does the assay work in vitro using purified components?

            The SLCA is typically performed using cell lysates, although it can also be adapted for live-cell assays. Firefly luciferase enzyme has been used in previous studies to show that cells can be co-transfected with both NLuc- and CLuc-tagged constructs or transfected separately and then combined. In our experiments, we commonly use HEK293T cells, which provide robust expression and very strong signal-to-noise ratios. While the assay has not been systematically tested across many different cell types, there is no inherent reason it would not work in other transfectable cell lines, as long as protein expression is adequate. The luciferase reaction requires both ATP and luciferin as substrates; ATP is not cell-permeable due to its hydrophilic nature whereas luciferin is cell-permeable. For this reason, most implementations of SLCA are performed in cell lysates, where ATP and luciferin can be freely accessed by the reconstituted luciferase enzyme. When using purified components, the assay functions similarly: purified NLuc- and CLuc-tagged proteins are combined in vitro, and bioluminescence is quantified after adding ATP and luciferin. This allows precise quantification of protein-protein interactions under controlled conditions.  The beauty and strength of performing the SLCA under cell-free conditions is that it is easily adaptable to 384-well format for high throughput screening of small molecule inhibitors, which we are now performing for HIV Rev SLCA.

2. It would be helpful to show some actual data obtained with the assay. Do the authors have some data to make a figure or perhaps get permission to use some published data from the assay. 

We have incorporated published data from the study “Cell-based and cell-free firefly luciferase complementation assay to quantify Human Immunodeficiency Virus type 1 Rev–Rev interaction” (Virology, Hansen et al., 2022) in the manuscript (specifically under the “SLCA Applications in Viruses” section, in the second paragraph of the subsection “Human Immunodeficiency Virus type 1 (HIV-1)”) to illustrate assay performance and robustness. As described in the text, the Rev-Rev interaction assay demonstrated a high Z’-factor of 0.85, confirming the reliability of the SLCA for both in vitro and in vivo applications and supporting its suitability for high-throughput screening of small-molecule inhibitors. In addition, our laboratory has obtained extremely strong preliminary SLCA data for Tat–Cyclin T1, HIV nucleocapsid, HIV capsid, and HCV core interactions. However, as these results remain unpublished, we will not include them in the current manuscript. It is hoped and expected that these data will be presented in future, peer-reviewed publications.

3. What amino acid residues constitute the A and B luciferase fragments? What are the sizes of the fragments. How was it decided where to split the protein?  

Firefly luciferase (~62 kDa) is commonly split as follows: Nluc (1-416 amino acids) and Cluc (398–550 amino acids). The split site was selected empirically to minimize spontaneous reconstitution in the absence of protein-protein interaction while maintaining structural stability of the fragments. Literature (e.g., Magliery et al., 2004; Lang et al., 2019) guides optimal split locations. The choice of fusion orientation with the proteins of interest is empirically determined to avoid steric hindrance or disruption of the interaction interface.

4. Table 1 has a column labeled “target”. This shows the protein interaction that is being targeted.  It does not state the actual target of the drug.  An additional column should be added that lists the target of the drug.

Thank you for bringing this to our attention. We added an additional column, “Target,” to specify the molecular target of each compound. We also removed immune checkpoint inhibitors such as Atezolizumab, Durvalumab, Nivolumab, and Pembrolizumab, as they are monoclonal antibodies targeting PD-1/PD-L1 pathways and are not small-molecule inhibitors. For the same reason, we also removed Blinatumomab from Table 1, as it is a bispecific monoclonal antibody (CD19/CD3) rather than a small-molecule inhibitor and thus falls outside the scope of this study.

5. The authors refer to a list of anti-cancer mAbs (Atezolizumab, etc.) as evidence that drugs can target protein:protein interactions. These are all mAbs.  Would the SLCA be suitable to find drugs that are mAbs?  Could the assay be used to screen for drugs against cell surface receptors?  It would be helpful to state which type of protein:protein interactions can be targeted. 

Yes, the SLCA can be used to identify monoclonal antibodies that disrupt protein-protein interactions. Unpublished work from Sutton laboratory demonstrate that a monoclonal antibody targeting the HIV-1 capsid effectively inhibits the capsid-capsid interaction in the assay, providing proof-of-concept for mAb screening. In fact, a separate McAb targeting HIV-1 capsid enhances the interaction!!  To date, we have not tested the SLCA for screening drugs against cell surface receptors. While the assay is primarily optimized for soluble proteins, it is most effective for cytosolic and nuclear protein-protein interactions. Extending SLCA to cell-surface receptor interactions remains an interesting possibility for future studies and could be explored with additional assay adaptations.

6.Some description of the enzyme substrates used in the assay should be added.

We appreciate the reviewer’s suggestion and have included a brief description of the enzyme substrates used in the assay. For firefly luciferase, the reaction involves D-luciferin, ATP, and Oâ‚‚, which are converted to oxyluciferin with the emission of light. In the case of Renilla luciferase, coelenterazine serves as the substrate and generates light without the need for ATP. The split GFP system does not have a substrate, whereas the split alkaline phosphatase assays have various substrates that are used to produce a colorimetric signal upon reconstitution of the enzyme. In vitro assays using cell lysates, the appropriate substrate is added for quantitative measurement of protein-protein interactions.

7. It would seem that the assay might be dependent on the site at which the A and B proteins are fused to the N-Luc and C-Luc fragments. Does it matter?  Does it always work?

The success of the split luciferase complementation assay (SLCA) can indeed depend on the site at which the A and B proteins are fused to the N-Luc and C-Luc fragments. While homotypic interactions such as capsid–capsid or Rev–Rev generally produce strong signals, heterotypic interactions are more empirical and depend largely on the size and structural properties of the proteins involved. Larger proteins may hinder proper folding or sterically block luciferase fragment reconstitution, reducing assay performance. For example, a Rev–CRM1 signal was not detected when full-length CRM1 (over 1,000 amino acids) was used, perhaps due to its large size. The orientation of the luciferase tag (N-terminal vs. C-terminal) can also influence folding and interface accessibility. Although the orientations can be empirically tested for both, it is not completely necessary as previous studies suggest that proteins in the 200–300 amino acid range generally perform well in the SLCA.  An interesting question is how large the proteins can be—we know the SLCA works incredibly well when the number of amino acids of each protein is <300.

.

8. A few minor grammatical errors to correct. Page 3 third paragraph should be “are broad” rather than “is broad”.  Also cite reference for the Rev:Rev assay screen in that paragraph.  Page 6 first paragraph should be “that have” not “that has”.

Thank you for bringing this to our attention. The grammatical errors have been corrected.

9. Has anyone tested LEN in SLCA? Does it work?  LEN might not inhibit CA:CA interaction but might stabilize it. 

Yes, we have tested LEN in the SLCA. Unpublished data from the Sutton lab show that LEN decreases the RLU of the HIV-1 capsid–capsid interaction by approximately 20-fold, indicating that LEN effectively inhibits the capsid–capsid interaction. Interestingly, LEN also disrupts an already formed CA-CA interaction in the SLCA, suggesting that the CA-CA interaction is ‘breathing.’  While the potential for LEN to stabilize certain interactions exists, these results support its inhibitory effect in the context of the SLCA. However, as these results remain unpublished, they cannot be included in the current manuscript for the review on SLCA.

Reviewer 2 Report

Comments and Suggestions for Authors

“Use of the split luciferase complementation assay to identify novel small molecules that disrupt essential protein-protein interactions of viruses” by Tisa Biswas and Richard E. Sutton is presented as a review, but its sturcture and content align more closely with a perspective piece. The inclusion of off topics dilutes the main focus, and the manuscript would benefit from a more focused and systematic approach. The authors should place greater emphasis on explaining the principles and applications of the split luciferase complementation assay , ideally supported by molecular-level illustrations of how the system functions in different biological contexts underlaying variations in use of luciferase sources. A big mistake is the lack of references to studies that have successfully employed SLCA to identify antiviral compounds. Including such examples would significantly strengthen the article’s relevance and scientific grounding.

A more detailed discussion of LEN targeting the HIV capsid would be particularly valuable. The manuscript should highlight LEN mechanism of action, using e.g. multiple high-resolution co-crystal structures available in the Protein Data Bank. Additionally, the impact of resistance mutations such as M66I, which render LEN inactive could be discussed with appropriate citations.

The introduction lacks sufficient referencing for several factual claims such as the interaction between Tat and Cyclin T1, and numerical data on viral infections, etc

. Expanding the introduction to include a brief overview of FDA-approved small molecule inhibitors for viral infections would also enhance its relevance and provide a stronger foundation for the discussion that follows (it would be better placed here). Moreover, the inclusion of immune checkpoint inhibitors such as Atezolizumab, Durvalumab, Nivolumab, and Pembrolizumab is puzzling, as these are monoclonal antibodies targeting PD-1/PD-L1 pathways and are not small molecule inhibitors and are not related to viral infections. Their relevance to the topic is unclear, and unless a compelling rationale is provided, their mention should be reconsidered. This also applies to other examples of FDA-approved therapeutics that are primarily monoclonal antibodies rather than small molecules.

The main text would benefit from more comprehensive referencing throughout, particularly when discussing specific compounds or mechanisms. For example, the statement regarding LEN’s inactivity against the M66I mutation should be supported by primary literature. Additionally, the inclusion of schematic figures illustrating viral cycles and the roles of specific viral proteins  would greatly help reader comprehension and should be presented for each mentioned virus.

while the topic is timely and potentially impactful, the manuscript requires substantial revision for a good a scientific review. A clearer focus on SLCA, improved referencing, inclusion of relevant case studies, and removal of unrelated content will significantly enhance the manuscript’s clarity, coherence, and scientific value.

Author Response

1. Use of the split luciferase complementation assay to identify novel small molecules that disrupt essential protein-protein interactions of viruses” by Tisa Biswas and Richard E. Sutton is presented as a review, but its structure and content align more closely with a perspective piece. The inclusion of off topics dilutes the main focus, and the manuscript would benefit from a more focused and systematic approach.

            We thank the reviewer for all the helpful comments and have implemented all the necessary comments to focus the manuscript on the SLCA system more towards virology.

2. The authors should place greater emphasis on explaining the principles and applications of the split luciferase complementation assay , ideally supported by molecular-level illustrations of how the system functions in different biological contexts underlaying variations in use of luciferase sources.

We thank the reviewer for this suggestion. We have placed greater emphasis on explaining the principles and applications of the split luciferase complementation assay (SLCA) in the revised manuscript, including detailed descriptions of how it functions in different biological contexts and the variations in luciferase sources. While we do not have structural information to provide molecular-level illustrations of SLCA, Figure 1 illustrates the assay principle and a general overview of its applications under different biological contexts. Unpublished data from capsid–capsid interactions (Sutton lab) further support the functional relevance of the assay, although these data are not included in the manuscript.

3. A big mistake is the lack of references to studies that have successfully employed SLCA to identify antiviral compounds. Including such examples would significantly strengthen the article’s relevance and scientific grounding.

We acknowledge the importance of including references to studies that have successfully employed SLCA to identify antiviral compounds. Currently, there are no published studies directly demonstrating SLCA-based identification of antiviral hits for the specific targets discussed in this manuscript. To provide a proof-of-concept, we cite a study using a related cell-based split luciferase complementation (SLC) assay in hepatitis B virus (HBV), where Arbidol and 20-deoxyingenol were identified from a 672-compound library as regulators of HBV core dimerization, which correspondingly altered HBV DNA replication in vitro (Wei et al., 2018). Although this approach differs from the classical SLCA used in HIV-1 and other viruses, it demonstrates that split luciferase complementation can be adapted to screen for inhibitors or modulators of critical viral protein-protein interactions across diverse viral systems. Unpublished data from the Sutton lab show that Roscovitine reduces HIV-1 Tat/cyclin T1 interaction as quantified by SLCA with an IC-50 value of ~10 mM.  The Sutton lab is NIH-funded and is currently performing a HTS using the Rev-Rev SLCA and has identified 2 small molecule hits of low mM potency (unpublished work), and Lenacapavir decreases RLU of the HIV-1 capsid–capsid interaction by approximately 20-fold, further supporting proof-of-concept that SLCA can identify inhibitory compounds. While these unpublished results cannot be included in the manuscript, it is hoped that future studies will validate and expand on these findings.

4. A more detailed discussion of LEN targeting the HIV capsid would be particularly valuable. The manuscript should highlight LEN mechanism of action, using e.g. multiple high-resolution co-crystal structures available in the Protein Data Bank. Additionally, the impact of resistance mutations such as M66I, which render LEN inactive could be discussed with appropriate citations.

We thank the reviewer for the suggestion regarding a more detailed discussion of LEN.  Although LEN serves as a relevant example of a small molecule that targets HIV-1 capsid-capsid interactions, the primary focus of this manuscript is on the SLCA and its applications rather than providing a comprehensive review of LEN. To address the comment, we have added a concise paragraph summarizing LEN’s mechanism of action: LEN binds at the interface of HIV-1 capsid hexamers, interfering with both assembly and disassembly of the viral core. This disruption prevents proper formation of the capsid and inhibits nuclear import of viral DNA. High-resolution co-crystal structure (Figure 4) was included to depict resistance mutations such as M66I. We also note that resistance mutations, such as M66I, can reduce LEN efficacy by altering the capsid interface, highlighting the ongoing need for identification of additional inhibitors. This revision emphasizes how SLCA can be applied to study capsid-targeting compounds, while maintaining the manuscript’s focus on the assay itself.  Parenthetically, in our HIV-1 CA-CA SLCA, we have tested a number of compounds related to LEN, provided by Dr. Stefan Sarafianos of Emory University, and they are all quite inhibitory—it is hoped that a manuscript will be submitted before the end of the year.

5. The introduction lacks sufficient referencing for several factual claims such as the interaction between Tat and Cyclin T1, and numerical data on viral infections, etc

We thank the reviewer for pointing this out. Additional references have been added throughout the introduction to support factual claims, including the interaction between HIV-1 Tat and Cyclin T1. All new citations are highlighted in yellow.

6. Expanding the introduction to include a brief overview of FDA-approved small molecule inhibitors for viral infections would also enhance its relevance and provide a stronger foundation for the discussion that follows (it would be better placed here).

We have expanded the introduction to include a brief overview of FDA-approved small molecule inhibitors for viral infections, such as Fostemsavir, Maraviroc, and Lenacapavir, highlighting how they target specific viral protein-protein interactions. This addition provides context for the broader discussion on SLCA by illustrating that small molecules can effectively disrupt viral protein complexes, thereby establishing the relevance and potential of using SLCA to identify new antiviral therapeutics.  Parenthetically, in our HIV clinic at VACT we do not use Fostemsavir or Maraviroc, and the VA has not yet approved the use of Lenacapavir for treatment due to its monetary expense, nor for prophylaxis.

7. Moreover, the inclusion of immune checkpoint inhibitors such as Atezolizumab, Durvalumab, Nivolumab, and Pembrolizumab is puzzling, as these are monoclonal antibodies targeting PD-1/PD-L1 pathways and are not small molecule inhibitors and are not related to viral infections. Their relevance to the topic is unclear, and unless a compelling rationale is provided, their mention should be reconsidered. This also applies to other examples of FDA-approved therapeutics that are primarily monoclonal antibodies rather than small molecules.

The examples of immune checkpoint inhibitors (Atezolizumab, Durvalumab, Nivolumab, and Pembrolizumab) and Blinatumomab have been removed from the section. The text has been revised to focus solely on FDA-approved small molecules that target protein-protein interactions, which better aligns with the scope and purpose of the manuscript.

8. The main text would benefit from more comprehensive referencing throughout, particularly when discussing specific compounds or mechanisms. For example, the statement regarding LEN’s inactivity against the M66I mutation should be supported by primary literature. Additionally, the inclusion of schematic figures illustrating viral cycles and the roles of specific viral proteins  would greatly help reader comprehension and should be presented for each mentioned virus.

Additional references have been incorporated throughout the main text to support statements regarding specific compounds and mechanisms, including LEN’s reduced efficacy against the M66I capsid mutation. While we agree that schematic figures can enhance visualization, we have chosen not to include additional life cycle illustrations in order to maintain the manuscript’s focus on the principles and applications of the SLCA, rather than detailed virological pathways. Instead, we have cited recent authoritative reviews for each virus that comprehensively describe their replication cycles and highlight the stages at which key viral proteins multimerize—for example, Dengue virus (Diamond & Pierson, 2015), Eastern equine encephalitis virus (Hasan et al., 2021), Western equine encephalitis virus (Gauci et al., 2009; Ma et al., 2025), Epstein–Barr virus (Chakravorty et al., 2022), and rabies virus (Schoehn et al., 2001; Nevers et al., 2022). These references provide detailed mechanistic and structural context for readers seeking additional background, ensuring scientific completeness without redundancy or excessive length. Additionally, Figure 5 has been included to illustrate the structural diversity of viral capsids across the six viruses discussed, thereby providing a concise visual overview that complements the text while preserving the manuscript’s central focus.

9. While the topic is timely and potentially impactful, the manuscript requires substantial revision for a good scientific review. A clearer focus on SLCA, improved referencing, inclusion of relevant case studies, and removal of unrelated content will significantly enhance the manuscript’s clarity, coherence, and scientific value.

We have carefully revised the manuscript to improve clarity, coherence, and scientific focus. Specifically, we have added missing references, a new figure (Figure 4), and refined our word choice and writing throughout the text to ensure that the manuscript no longer reads as a perspective piece but rather as a structured, evidence-based scientific review. We also removed the five drug examples from Table 1 and added a new column labelled “target” to Table 1 to maintain a focused discussion on the SLCA and its applications in studying and identifying inhibitors of viral protein–protein interactions. Furthermore, we provided a clear rationale for retaining the FDA-approved compounds section, emphasizing its importance in illustrating how SLCA findings can be translated into therapeutic development pipelines. This section now serves to contextualize the potential of SLCA within the broader framework of antiviral drug discovery for the six different viruses.  All the changes are highlighted in yellow.)

Round 2

Reviewer 1 Report

Comments and Suggestions for Authors

This is a useful review; well written with clear figures.  Minor revisions have improved the resubmitted article. 

One point worthy of addressing is which types of interactions are best suited for the SLCA rather than other assays.  For example, maraviroc, targeting CCR5, was not identified with such an assay and SLCA would not have be useful to screen for coreceptor antagonists.    

Author Response

Comments: This is a useful review; well written with clear figures.  Minor revisions have improved the resubmitted article. One point worthy of addressing is which types of interactions are best suited for the SLCA rather than other assays.  For example, maraviroc, targeting CCR5, was not identified with such an assay and SLCA would not have be useful to screen for coreceptor antagonists.   

Our response: We thank the reviewer for the positive feedback and for highlighting this important point. The SLCA is indeed best suited for detecting intracellular protein-protein interactions, where both fusion partners can be co-localized within the same cellular compartment. However, interactions involving extracellular or membrane-bound targets, such as CCR5 and its antagonists, have not yet been assessed using the SLCA and could be investigated in the future. We have clarified this distinction in the manuscript to more accurately reflect the SLCA’s appropriate applications. We would also like to note that we modified figure 2 and 6 to avoid any copyright issues or reuse concerns.

Reviewer 2 Report

Comments and Suggestions for Authors

Dear Authors,
While I appreciate the changes made to the manuscript, this is not a review paper which by the definition analyzes existing and reported research in the literature. As I mentioned in my previous comments regarding the need for references, the current manuscript is more aligned with a perspective article. In its present form, it can only be accepted as such, therefore I suggest changing the type of publication to "perspective" or to at least wait for the other publication to be released. However, a review article of a technique based on one publication from the same authors is also not well-regarded.

Author Response

Comments: Dear Authors, While I appreciate the changes made to the manuscript, this is not a review paper which by the definition analyzes existing and reported research in the literature. As I mentioned in my previous comments regarding the need for references, the current manuscript is more aligned with a perspective article. In its present form, it can only be accepted as such, therefore I suggest changing the type of publication to "perspective" or to at least wait for the other publication to be released. However, a review article of a technique based on one publication from the same authors is also not well-regarded.

Our response:  We thank the reviewer for their thoughtful assessment and acknowledge their concern regarding the classification of the manuscript. Our intention was to provide a focused synthesis of the applications of the split luciferase complementation assay across multiple viral models, drawing from the existing body of literature on split enzymatic techniques, which have been broadly implemented in diverse biological contexts. While our own primary publication contributes to this area, the manuscript was developed using previously published studies employing split luciferase and related split systems, rather than relying solely on our work. These references employed split luciferase: Wei et al., 2018; Villalobos et al., 2008; Paulmurugan et al., 2005; Liang et al., 2022; Chen et al., 2022. The research articles used split systems such as split GFP system and split alkaline phosphatase assay  (Magliery et al., 2005; Shen et al., 2023). The split enzymatic system has been shown to be a fantastic method in analyzing and quantifying protein-protein interactions under specific conditions.

We fully respect the reviewer’s perspective that the current structure may align more closely with a perspective article. We defer to your judgment regarding the most appropriate article type and are willing to revise the manuscript’s classification accordingly.

Although we received high scores across all categories from the first reviewer who did not suggest revisions in any area, we did observe the lower scores in these categories from Reviewer 2 but did not see specific comments indicating how we might improve the manuscript. We greatly appreciate any clarification or guidance on how to strengthen these sections so we can revise the manuscript accordingly. We would also like to point out that we modified figure 2 and 6 to avoid any copyright or reuse concerns.

Thank you for your attention and I look forward to a favorable response.

Round 3

Reviewer 2 Report

Comments and Suggestions for Authors

the manuscript biomolecules-3923528 is now perfectly fine as a perspective. The changes that authors made address all of my previous concerns. I found only few minor editorial issues mentioned below. Otherwise, the manuscript is ready for the acceptance. 

I think this is a duplication

 „The SLCA specifically uses firefly luciferase, which is divided into two inactive fragments:

the N-terminal (Nluc) and the C-terminal (Cluc). The firefly luciferase enzyme can

be successfully bisected as follows: Nluc (1-416 amino acids) and Cluc (398-500 amino acids)

(Lang et al., 2019).”

On page 8 figure from PDB:8GDV is copied twice (one should be deleted)

Figure 5 is not mentioned in the text.

Author Response

We thank the reviewer for their feedback and comments. All the errors were corrected in the final draft. Figure 5 is mentioned under sections 2.3-2.6.